# Proteogenomic landscape and clinical characterization of GH-producing pituitary adenomas/somatotroph pituitary neuroendocrine tumors

Azusa Yamato[1,14], Hidekazu Nagano[1,14], Yue Gao[1,2], Tatsuma Matsuda[1,2], Naoko Hashimoto[1,3], Akitoshi Nakayama[1], Kazuyuki Yamagata [1], Masataka Yokoyama[1], Yingbo Gong[1], Xiaoyan Shi[1], Siti Nurul Zhahara[1], Takashi Kono[1], Yuki Taki[1], Naoto Furuki[1], Motoi Nishimura[4], Kentaro Horiguchi [2], Yasuo Iwadate [2], Masaki Fukuyo[5], Bahityar Rahmutulla [5], Atsushi Kaneda [5], Yoshinori Hasegawa[6], Yusuke Kawashima [6], Osamu Ohara [6], Tetsuo Ishikawa [7,8,9], Eiryo Kawakami[7,8], Yasuhiro Nakamura[10], Naoko Inoshita[11], Shozo Yamada[12], Noriaki Fukuhara[13], Hiroshi Nishioka[13] & Tomoaki Tanaka [1,3✉]

The clinical characteristics of growth hormone (GH)-producing pituitary adenomas/somatotroph pituitary neuroendocrine tumors (GHomas/somatotroph PitNETs) vary across patients. In this study, we aimed to integrate the genetic alterations, protein expression profiles, transcriptomes, and clinical characteristics of GHomas/somatotroph PitNETs to identify molecules associated with acromegaly characteristics. Targeted capture sequencing and copy number analysis of 36 genes and nontargeted proteomics analysis were performed on fresh-frozen samples from 121 sporadic GHomas/somatotroph PitNETs. Targeted capture sequencing revealed *GNAS* as the only driver gene, as previously reported. Classification by consensus clustering using both RNA sequencing and proteomics revealed many similarities between the proteome and the transcriptome. Gene ontology analysis was performed for differentially expressed proteins between wild-type and mutant *GNAS* samples identified by nontargeted proteomics and involved in G protein–coupled receptor (GPCR) pathways. The results suggested that *GNAS* mutations impact endocrinological features in acromegaly through GPCR pathway induction. ATP2A2 and ARID5B correlated with the GH change rate in the octreotide loading test, and WWC3, SERINC1, and ZFAND3 correlated with the tumor volume change rate after somatostatin analog treatment. These results identified a biological connection between *GNAS* mutations and the clinical and biochemical characteristics of acromegaly, revealing molecules associated with acromegaly that may affect medical treatment efficacy.

[1] Department of Molecular Diagnosis, Graduate School of Medicine, Chiba University, Chiba, Japan. [2] Department of Neurological Surgery, Chiba University Hospital, Chiba, Japan. [3] Research Institute of Disaster Medicine, Chiba University, Chiba, Japan. [4] Division of Laboratory Medicine, Chiba University Hospital, Chiba, Japan. [5] Department of Molecular Oncology, Graduate School of Medicine, Chiba University, Chiba, Japan. [6] Department of Applied Genomics, Kazusa DNA Research Institute, Chiba, Japan. [7] Advanced Data Science Project, RIKEN Information R&D and Strategy Headquarters, RIKEN, Kanagawa, Japan. [8] Artificial Intelligence Medicine, Graduate School of Medicine, Chiba University, Chiba, Japan. [9] Department of Extended Intelligence for Medicine, The Ishii-Ishibashi Laboratory, Keio University School of Medicine, Tokyo, Japan. [10] Division of Pathology, Faculty of Medicine, Tohoku Medical and Pharmaceutical University, Miyagi, Japan. [11] Department of Pathology, Moriyama Memorial Hospital, Tokyo, Japan. [12] Hypothalamic and Pituitary Center, Moriyama Neurological Center Hospital, Tokyo, Japan. [13] Department of Hypothalamic and Pituitary Surgery, Toranomon Hospital, Tokyo, Japan. [14] These authors contributed equally: Azusa Yamato, Hidekazu Nagano. ✉email: tomoaki@restaff.chiba-u.jp

COMMUNICATIONS BIOLOGY | (2022)5:1304 | https://doi.org/10.1038/s42003-022-04272-1 | www.nature.com/commsbio 1

Pituitary adenomas/pituitary neuroendocrine tumors (pituitary adenomas/PitNETs) account for ~15% of all primary intracranial neoplasms. Among them, growth hormone (GH)-producing pituitary adenomas/somatotroph pituitary neuroendocrine tumors (GHomas/somatotroph PitNETs) are the second most common hormone-producing adenomas after prolactin-producing adenomas[1,2]. Acromegaly, due to GHomas/somatotroph PitNETs, presents with a wide variety of symptoms, such as the enlargement of the distal extremities and tongue, cardiac hypertrophy, osteoarthropathy, metabolic disorders, and malignancy. These complications can lead to a high mortality rate if GH overproduction remains uncontrolled. The first-line treatment for GHomas/somatotroph PitNETs is surgical tumor resection. Medical treatments, including somatostatin analog (SSA) or GH receptor antagonists, may be used in cases of noncurative resection or for preoperative tumor volume reduction. SSAs have efficacy in suppressing the secretion of GH and shrinking tumors when they bind to the somatostatin receptor on the surface of the tumor cell membrane[3]. Disappointingly, some patients do not achieve remission, regardless of the treatment strategy. Therefore, the elucidation of the underlying biological mechanisms may identify targets with more effective treatment effects.

Somatic mutations in the α-subunit of the stimulatory Gs protein (GNAS) gene, resulting in the constitutive activation of cAMP, are observed in 30–50% of patients with acromegaly[4]. Although whole-exome analyses of GHomas/somatotroph PitNETs revealed that GNAS was the only driver gene of GHomas/somatotroph PitNETs[1,2], previous reports have identified other causative gene mutations. Germline mutations in aryl hydrocarbon receptor-interacting protein (AIP), multiple endocrine neoplasia type 1 (MEN1), and cyclin-dependent kinase inhibitor 1B (CDKN1B) have been reported in a small percentage of young patients with acromegaly[5–9] and germline or somatic microduplications of the chromosome region Xq 26.3, which contains the gene encoding the G protein–coupled receptor 101 (GPR101), have been implicated in younger-onset gigantism[10]. Previous reports examining genotype–phenotype relationships in patients with acromegaly showed that patients with GNAS mutations (GNAS-MT) presented with smaller tumors and had better SSA responsiveness than those without GNAS-MT[11,12], although the relationships between other gene mutations and clinical features remain unclear. Several reports have presented correlations between somatostatin receptor 2 (SSTR2) mRNA expression levels and the efficacy of SSAs[3,13,14]. However, these findings do not sufficiently explain the various clinical characteristics of acromegaly.

Accumulating evidence shows the involvement of genetic events other than single mutations or small indels in pituitary adenomas/PitNETs. Several studies have identified high levels of genomic instability and a high frequency of copy number alterations (CNAs) in pituitary adenomas/PitNETs[15–20], and genomic CNAs may be especially frequent in GHomas/somatotroph PitNETs[2,21]. An association between DNA damage and cAMP activation has also been indicated[19]. Epigenetic alterations in GHomas/somatotroph PitNETs have also been reported[19,22], and multiomics analyses have revealed potential alterations in gene expression patterns in GH-producing adenomas. Salmon et al. reported an important relationship between DNA methylation patterns and gene expression profiles in pituitary adenomas/PitNETs[22], and Neou et al. presented molecular profiles of pituitary adenomas/PitNETs based on the performance of various omics analyses[23].

Although recent studies of pituitary adenomas/PitNETs have aimed to clarify the underlying tumorigenesis mechanisms and disease etiologies, the association between genetic alterations and clinical characteristics remains unclear, and the factors that affect responsiveness to treatment are unknown. In addition, few genetic studies have been limited to GHomas/somatotroph PitNETs. Here, we performed gene alteration analysis and proteomics analysis on 121 GHomas/somatotroph PitNETs and integrated these results with the clinical characteristics of acromegaly. We attempted to identify key players involved in shaping the clinical features of acromegaly, especially those related to treatment efficacy. Our study revealed the importance of GNAS-MT in terms of clinical and biochemical characteristics and identified molecules that may be involved in the responsiveness to medical treatment.

**Table 1 Clinical characteristics of GH-producing pituitary adenoma/ somatotroph pituitary neuroendocrine tumor patients and comparison of those with wild-type versus mutant GNAS.**

|  | All patients (n = 121) | GNAS wild-type (n = 52) | GNAS mutant (n = 69) | p value |
|---|---|---|---|---|
| Clinical characteristics |  |  |  |  |
| Age (years) | 48.0 (39.0–59.5) | 46.5 (40.0–60.5) | 49.0 (38.5–59.5) | 0.763 |
| Sex, n (male/female) | 51/70 | 20/32 | 31/38 | 0.577 |
| Basal GH(ng/mL) | 15.7 (7.2–34.1) | 13.4 (4.6–31.3) | 16.9 (8.3–48.5) | 0.061 |
| IGF-1 (ng/mL) | 630.0 (473.0–817.5) | 602.0 (448.0–829.3) | 237.5 (505.0–817.5) | 0.745 |
| IGF-1 SDS | 7.0 ± 2.4 | 7.2 ± 2.6 | 6.9 ± 2.3 | 0.580 |
| PRL (ng/mL) | 13.4 (8.1–29.1) | 10.3 (7.4–21.0) | 20.2 (9.8–37.3) | 0.005 |
| GH change by octreotide test (%) | −87.7 (−93.9-−61.3) | −83.4 (−92.0-−57.1) | −89.6 (−95.0-−69.6) | 0.063 |
| GH change by bromocriptine test (%) | −65.6 (−83.8-−23.7) | −39.3 (−68.0-−7.3) | −79.1 (−90.2-−55.4) | <0.001 |
| Tumor volume (mm³) | 1633.1 (651.9–4412.4) | 2323.7 (1020.3–4990.0) | 1135.4 (406.2–3493.1) | 0.023 |
| Knosp grade 0–2/3–4, n | 93/28 | 33/19 | 60/9 | 0.004 |
| preoperative therapy |  |  |  |  |
| preoperative SSA treatment, n (%) | 67 (55.4%) | 33 (63.5%) | 34 (49.3%) |  |
| GH change by preoperative SSA (%) | −78.9 (−90.8-−48.1) | −55.5 (−80.8-−41.7) | −83.2 (−92.7-−64.2) | 0.007 |
| Tumor volume change by preoperative SSA (%) | −23.1 ± 21.3 | −19.9 ± 18.6 | −26.2 ± 23.5 | 0.237 |
| postoperative profile |  |  |  |  |
| Nadir GH after OGTT (ng/mL) | 0.42 (0.27–0.84) | 0.50 (0.26–1.0) | 0.39 (0.27–0.67) | 0.514 |
| IGF-1 (ng/mL) | 189.0 (137.0–227.0) | 186.5 (153.0–243.2) | 190.0 (130.0–220.0) | 0.237 |
| IGF-1 SDS | 0.7 ± 1.6 | 0.9 ± 1.7 | 0.5 ± 1.5 | 0.119 |
| postoperative treatment, n (%) | 21 (17.4%) | 9 (17.3%) | 12 (17.4%) | 0.990 |

GH Growth hormone, PRL Prolactine, SSA Somatostatin analog, OGTT Oral glucose tolerance test.

## Results

### Mutational landscape assessment by targeted capture sequencing in GHomas/somatotroph PitNETs.
A summary of the clinical data for all 121 GHomas/somatotroph PitNETS patients is shown in Table 1. The mean basal GH level was high (15.7 ng/mL, IQR; 7.2–34.1), and the average IGF-1 SD score was 7.0 ± 2.4 (SD). The mean tumor volume was 1633.1 mm³ (IQR; 651.9–4412.4), and 23.1% of patients presented with a high Knosp grade (≥3). All 121 patients underwent transsphenoidal surgery (TSS), and 67 patients (55.4%) received preoperative SSA treatment. We evaluated biological remission by postoperative oral glucose tolerance test (OGTT) or normalization of IGF-1 SDS, resulting in 21 patients requiring additional therapy. The clinical data for all patients are provided in Supplementary Data 1. We performed targeted capture sequencing (TCS) of 36 genes (Supplementary Data 2) in all tumor tissue samples. The average sequencing depth was 1838.3 (range 29×–11973×), and 99.7% of the target regions were covered at least 50×. We detected a total of 83 mutations in 30 genes, with a median of 3.0 somatic coding region variants per tumor (range, 0–9). Among these mutations, 37 mutations in 18 genes were defined as recurrent (Fig. 1a).

We observed various activating GNAS-MT (57.0% of patients) at known hot spots in 69 patients in our cohort; p.Arg201Cys was identified in 45 patients, p.Arg201His in 5 patients, p.Arg201Ser in 3 patients, p.Gln227Leu in 13 patients, p.Gln227Arg in 2 patients, and p.Gln227Glu in 1 patient. One patient carried both p.Arg201Cys and p.Gln870Leu mutations. We also detected 3 unreported GNAS-MT (p.Gly49Arg, p.Ser111Asn, and p.Ala249Asp). The number of patients was one. We mapped the mutations to the crystal structure of the Gsα protein (PDB: 6AU6) and found that p.Gly49 is located in proximity to the GTP binding domain (Supplementary Fig. 1a, b).

We detected several gene mutations that have previously been reported in GHomas/somatotroph PitNETs [AIP, GPR101, somatostatin receptor 5 (SSTR5), arrestin beta 1/2(ARRB1/2)] and associated with the cAMP pathway [protein kinase cAMP-activated catalytic subunit alpha (PRKACA), protein kinase cAMP-activated catalytic subunit beta (PRKACB), and protein kinase cAMP-dependent type I regulatory subunit alpha (PRKAR1A)]. AIP mutations were observed in 8 patients, and a GPR101 mutation was detected in 1 patient. SSTR5 was mutated in 3 patients. We detected 2 patients with ARRB1 mutations and 2 patients carrying ARRB2 mutations. PRKACA mutation was detected in 1 patient, and PRKAR1A and PRKACB mutations were observed in 2 patients each. Most of these mutations were not recurrent in our study, and no hotspots have previously been reported for these genes.

### Copy number analysis revealed the loss of SDHx.
We calculated copy numbers from the TCS data using CNVkit, as described in the Methods, and detected 21 cases with CNAs (Fig. 1a). Fourteen patients showed copy number gains, and 10 patients presented copy number losses. Seven patients harbored CNAs in more than 2 genes (range, 1–5). Notably, among the 14 patients with identified copy number gains, 7 patients harbored gains in SSTR5, 5 patients harbored gains in GPR101, and one patient presented CNAs in both genes. Among patients with identified copy number losses, 6 patients harbored losses in PRKACB, and 4 patients presented losses in succinate dehydrogenase complex (SDHx) genes, including SDHB, SDHC, and SDHD.

### Clinical characteristics of patients with GNAS mutations.
Although we identified 37 recurrent mutations in this study, we focused specifically on GNAS-MT. GNAS mutations (GNAS-MT) are well-known driver mutations of GHomas/somatotroph Pit-NETs, and more than half of the patients in our study harbored these mutations. A comparison between patients with GNAS-MT and those without mutations (GNAS-WT) is shown in Table 1 and Fig. 1b, c. The GNAS-MT group presented significantly better responsiveness to bromocriptine, consistent with a previous report that GNAS-MT adenomas are associated with higher dopamine receptor 2 mRNA expression than GNAS-WT adenomas[23]. A bubble plot analysis indicated that the GNAS-MT group also presented with smaller tumors and lower Knosp grades, although each dataset was associated with an extremely wide range. The GNAS-MT group tended to present with higher basal plasma GH levels and better responsiveness to octreotide loading test than the GNAS-WT group. The GNAS-MT group also showed a significantly better GH change rate following preoperative SSA treatment than the GNAS-WT group. However, the percentage of patients who required postoperative therapy was not significantly different between the two groups.

### Consensus clustering-based transomics classification of pituitary adenomas/PitNETs.
We performed proteomics and RNA sequencing analyses in pituitary adenomas/PitNETs (45 non-functioning pituitary adenomas/non-functioning pituitary neuroendocrine tumors [NFPAs/non-functioning PitNETs] for comparison, 60 GHomas/somatotroph PitNETs among 121 cases of the TCS cohort). The clinical data for all NFPAs/non-functioning PitNETs patients are provided in Supplementary Data 3. First, to clarify the transomics analysis, we examined the correlation between RNA and protein expression from RNA sequencing and proteomics data. Figure 2a shows a scatter plot of the correlation coefficient of a significantly positive correlation gene, with a mean Spearman's correlation coefficient of 0.476 (Fig. 2a). This is similar to the results of the correlation analysis of RNA sequencing and proteomics in human rectal colon cancer[24] and may represent tumor characteristics in pituitary tumors. Next, the whole gene was analyzed. Although 89.1% of the genes showed a positive mRNA-protein correlation, only 65.2% were significantly correlated (Fig. 2b). The average Spearman's correlation between mRNA and protein variations was 0.330. There were uncorrelated genes and negatively correlated genes. These results suggest that there are networks that could only be identified by transomics. To test whether the concordance between protein and mRNA variation was related to the biological function of the gene product, we performed Kyoto Encyclopedia of Genes and Genomes enrichment analysis. Genes involved in several hypothalamic-pituitary-adrenal axis signaling pathways showed concordant mRNA and protein variations (Fig. 2c). These results suggest that the transomics data also accurately represented pituitary tumor characteristics. Consensus clustering subtyping was applied to RNA sequencing datasets, proteomics datasets, and the combination of these two datasets (defined as the transomics data set), exploring between 2 and 8 K-means clusters. Consensus cumulative distribution functions (CDF) and delta area (change in the CDF area) plots were generated for each dataset to determine the optimal K value (Supplementary Fig. 2). The Sankey diagram depicts the flow of cluster assignments for NFPAs/non-functioning PitNETs and GHomas/somatotroph PitNETs across data types. To clarify the significance of performing transomics analyses, we examined the classification by consensus clustering using both RNA sequencing and proteomics data. The appropriate cluster sizes were determined to be K = 5 for RNA sequencing data, K = 4 for proteomics data, and K = 4 for the transomics combination (Fig. 2d). The results showed that transomics analysis better reflects protein expression classification compared to RNA analysis.

RNA sequencing classification may differ from proteomics classification because the expression of RNA and protein molecules is not always correlated (Fig. 2a, b). However, similar expression patterns were identified for pituitary-specific transcription factors

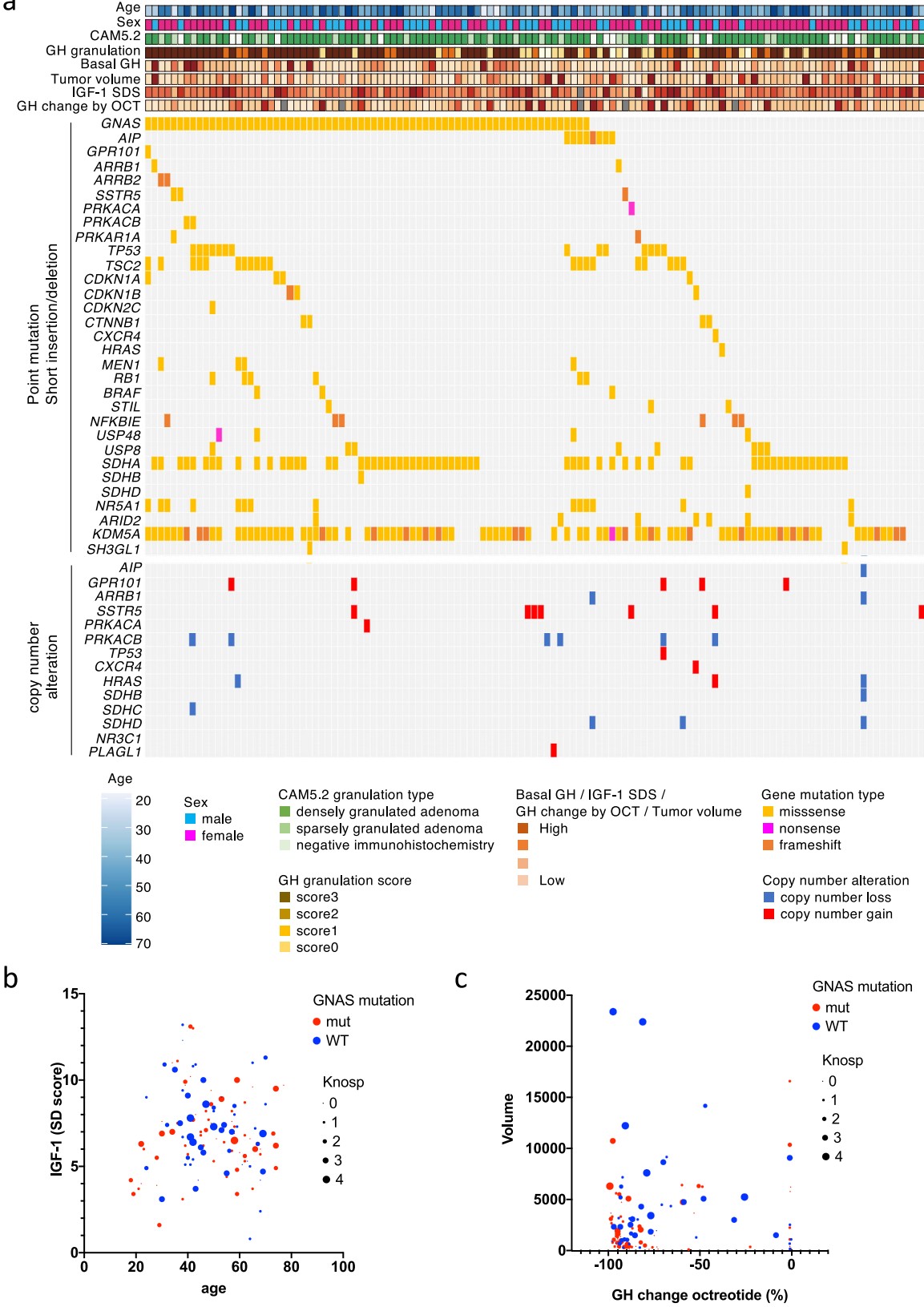

**Fig. 1 The landscape of gene mutations and CNAs in GHomas/somatotroph PitNETs. a** The landscape of gene mutations and copy number alterations (CNAs) in 121 growth hormone (GH)-producing pituitary adenomas/somatotroph PitNETs obtained by targeted capture sequencing (TCS), together with clinical and pathological annotations. **b** Bubble plot analysis, with age as the x-axis and the IGF-1 SD score as the y-axis. Blue circles represent *GNAS* wild-type, and red circles represent *GNAS* mutations. The circle size is relative to the Knosp grade. **c** Bubble plot analysis with GH change after octreotide treatment as the x-axis and tumor volume as the y-axis. Blue circles represent wild-type, and red circles represent mutations. The circle size is relative to the Knosp grade.

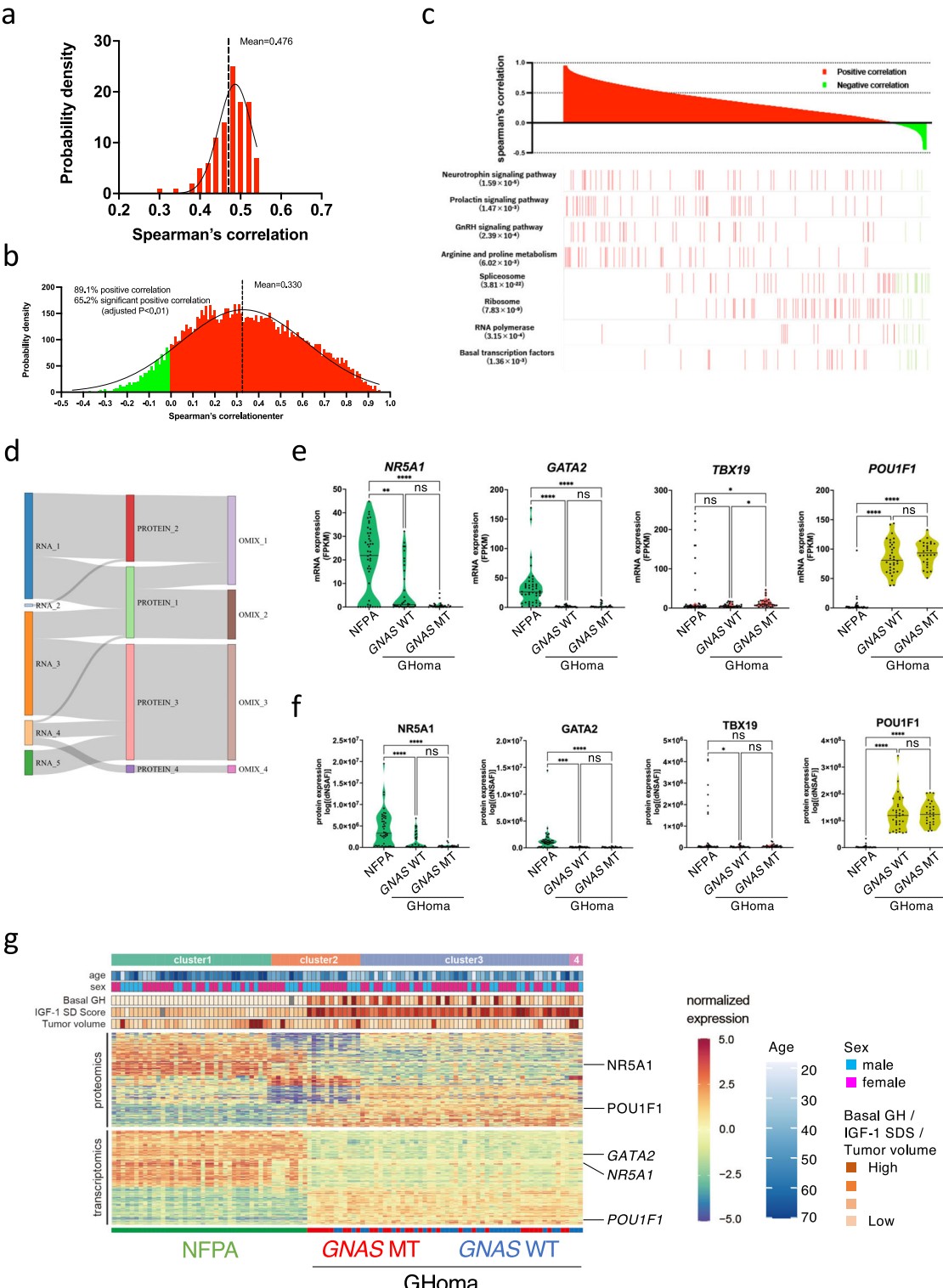

in the classification of NFPAs/non-functioning PitNETs and GHomas/somatotroph PitNETs (Fig. 2e, f). Although some differences were observed in the classification between the proteomics and transomics analyses, many similarities were also identified (Fig. 2d). Transomics analysis suggests that there may be molecules that exhibit variation in only protein expression that reflect the characteristics of pituitary adenomas/PitNETS to the best of our knowledge.

To explore the intrinsic cohort structure using the full complement of single-omics data, clustering was performed for mRNA and protein using non-negative matrix factorization (NMF)[25], which yielded between 2 and 4 clusters for the single-omics analyses (Fig. 2g). A GO analysis of a group of molecules with significantly higher expression in Cluster 3 by proteomics revealed the terms GH synthesis, secretion and action (Supplementary Data 4). Transcriptomic data were only able to distinguish between NFPAs/non-functioning PitNETs and GHomas/somatotroph PitNETs, whereas proteomics data were able to distinguish among GHomas/somatotroph PitNETS with and without GNAS mutation. Therefore, we focused on both

**Fig. 2 Consensus clustering-based transomics classification of pituitary adenomas/PitNETs. a** Steady-state mRNA and protein abundances were positively correlated with a mean Spearman's correlation coefficient of 0.476. **b** mRNA and protein variations were positively correlated for most (89.1%) mRNA-protein pairs and 65.2% of mRNA-protein pairs showed significant correlations, with a mean Spearman's correlation coefficient of 0.330. Negative correlations are shown in green and positive correlations in red. **c** mRNA and protein levels displayed dramatically different correlations for genes involved in different biological processes. Red and green indicate positive and negative correlations, respectively. **d** Results of unsupervised, nonnegative matrix factorization (NMF) subtyping applied to individual data types. The Sankey diagram depicts the flow of cluster assignments. **e** Violin plot depicting the mRNA expression of *NR5A1*, *GATA2*, and *TBX19* in NFPA and GHoma. **f** Violin plot depicting the protein expression of nuclear receptor subfamily 5 group A member 1 (NR5A1), GATA-binding factor 2 (GATA2), and T-box transcription factor 19 (TBX19) in NFPA and GHoma. **g** Unsupervised multiomics subtyping via NMF identified four molecular subtypes with distinct multiomics expression patterns. NFPA: nonfunctional pituitary adenoma/non-functioning pituitary neuroendocrine tumor (PitNETs), GHoma: GH-producing pituitary adenoma/somatotroph Pituitary neuroendocrine tumor (PitNETs). $*p < 0.05$, $**p < 0.01$, $***p < 0.001$, $****p < 0.0001$ by 1-way ANOVA. $n = 44$ in NFPA, $n = 34$ in GHoma with *GNAS* WT, $n = 28$ in GHoma with *GNAS* MT.

proteomics and genomics approaches for the characterization of GHomas/somatotroph PitNETs.

**Proteogenomic characterization of nonfunctioning and GHomas/somatotroph PitNETs.** The samples in which the proteins could not be identified by proteomics analysis were excluded. The protein expression of key transcription factors was compared between NFPAs/non-functioning PitNETs and GHomas/somatotroph PitNETs using uniform manifold approximation and projection (UMAP) analysis (Supplementary Fig. 3). The results showed that NFPAs/non-functioning PitNETs and GHomas/somatotroph PitNETs could be divided into 2 distinct groups, likely because NFPAs/non-functioning PitNETs and GHomas/somatotroph PitNETs have distinctly different protein expression characteristics. However, among GHomas/somatotroph PitNETs, no clear differences in protein expression characteristics were identified between cases with and without *GNAS*-MT. The expression of key transcription factors that contributed to pituitary differentiation visualized using a feature plot, revealed that nuclear receptor subfamily 5 Group A member 1 (NR5A1) and GATA-binding protein 3 (GATA3) are expressed at high levels in most NFPAs/non-functioning PitNETs samples, with some GHomas/somatotroph PitNETs also demonstrating high expression levels. In contrast, POU class 1 homeobox 1 (POU1F1), a key regulatory factor, was highly expressed in GHomas/somatotroph PitNETs, whereas POU1F1 and GH were not expressed in NFPAs/non-functioning PitNETs. SSTR5 was expressed heterogeneously in most GHomas/somatotroph PitNETs, with some expression observed in NFPAs/non-functioning PitNETs (Fig. 3a).

The Brunner–Munzel test was used to quantitatively compare protein expression levels between NFPAs/non-functioning PitNETs and GH-producing adenomas and between *GNAS*-WT and *GNAS*-MT GH-producing adenomas. A total of 7300 differentially expressed molecules were identified between NFPAs/non-functioning PitNETs and GHomas/somatotroph PitNETs, and 714 differentially expressed molecules were identified between *GNAS*-WT and *GNAS*-MT GH-producing adenomas. To evaluate the protein expression profiles of the 458 molecules that were identified as significantly differentially expressed in both comparisons (Fig. 3b, c), gene ontology (GO) analysis was performed. GO analysis (http://geneontology.org) revealed that the *GNAS* mutation influenced several binding functions and GPCR pathways (Supplementary Data 5). Considering these results collectively, we hypothesize that *GNAS*-MT induce GPCR pathways in GHomas/somatotroph PitNETs, influencing endocrinological characteristics, such as the response to drug treatments.

**Correlation between the protein expression profile and clinical characteristics of Acromegaly.** We first evaluated the expression of some characteristic proteins by pathological analysis. In our study, the immunohistochemistry (IHC) score for SSTR2, which

is known to serve as the primary SSA receptor in GHomas/somatotroph PitNETs, did not differ between the *GNAS*-WT and *GNAS*-MT groups (Fig. 4a). Similarly, the IHC scores for GH and the CAM5.2 cytokeratin IHC pattern did not differ between the two groups (Fig. 4b, c).

Based on the results of the nontargeted proteomics analysis, we compared protein expression levels between three groups: NFPAs/non-functioning PitNETs, *GNAS*-WT GHomas/somatotroph PitNETs, and *GNAS*-MT GHomas/somatotroph PitNETs. The protein expression level of T-box transcription factor 19 (TBX19, also known as TPIT) did not differ among the 3 groups, whereas the expression level of POU1F1 was significantly higher in the *GNAS*-WT and *GNAS*-MT groups than in the NFPAs/non-functioning PitNETs group (Fig. 4d, e), consistent with the regulation of transcription factors during pituitary development. The protein expression levels of AIP were significantly higher in the NFPAs/non-functioning PitNETs group than in the GHomas/somatotroph PitNETs group (Fig. 4f). Conversely, SSTR2 expression was significantly higher in the *GNAS*-WT and *GNAS*-MT groups than in the NFPAs/non-functioning PitNETs group (Fig. 4g).

Next, we focused on 4 molecules: SSTR5, sigma nonopioid intracellular receptor 1 (SIGMAR1), adhesion G protein–coupled receptor V1 (ADGRV1), and sortilin-related VPS10 domain–containing receptor 3 (SORCS3). These molecules are involved in GPCR activity and were among the 458 identified differentially expressed proteins. The expression levels of these molecules were significantly different between the NFPAs/non-functioning PitNETs and GHomas/somatotroph PitNETs groups: SSTR5 expression was higher in the GHomas/somatotroph PitNETs group than in the NFPAs/non-functioning PitNETs group, and there was no difference in expression with or without *GNAS*-MT (Fig. 4h). The SSTR2/5 protein expression ratio was not significantly different between the NFPAs/non-functioning PitNETs and GHomas/somatotroph PitNETs groups, nor were they expressed with or without *GNAS*-MT in the GHomas/somatotroph PitNETs group (Fig. 4i). SIGMAR1 expression was lower in the GHomas/somatotroph PitNETs group than in the NFPAs/non-functioning PitNETs group and was lower in the *GNAS*-MT groups than in the *GNAS*-WT group (Fig. 4j). ADGRV1 expression was not significantly different between the NFPAs/non-functioning PitNETs and GHomas/somatotroph PitNETs groups, nor was it different with or without *GNAS*-MT in the GHomas/somatotroph PitNETs group (Fig. 4k). SORCS3 expression was higher in GHomas/somatotroph PitNETs *GNAS*-WT than in NFPAs/non-functioning PitNETs and lower in *GNAS*-MT expression than in *GNAS*-WT (Fig. 4l). To determine whether these GPCR-related molecules affect the clinical characteristics of acromegaly, we analyzed the correlations between the expression levels of these four proteins and clinical characteristics, including the GH change rate following the octreotide loading test and the tumor volume change rate

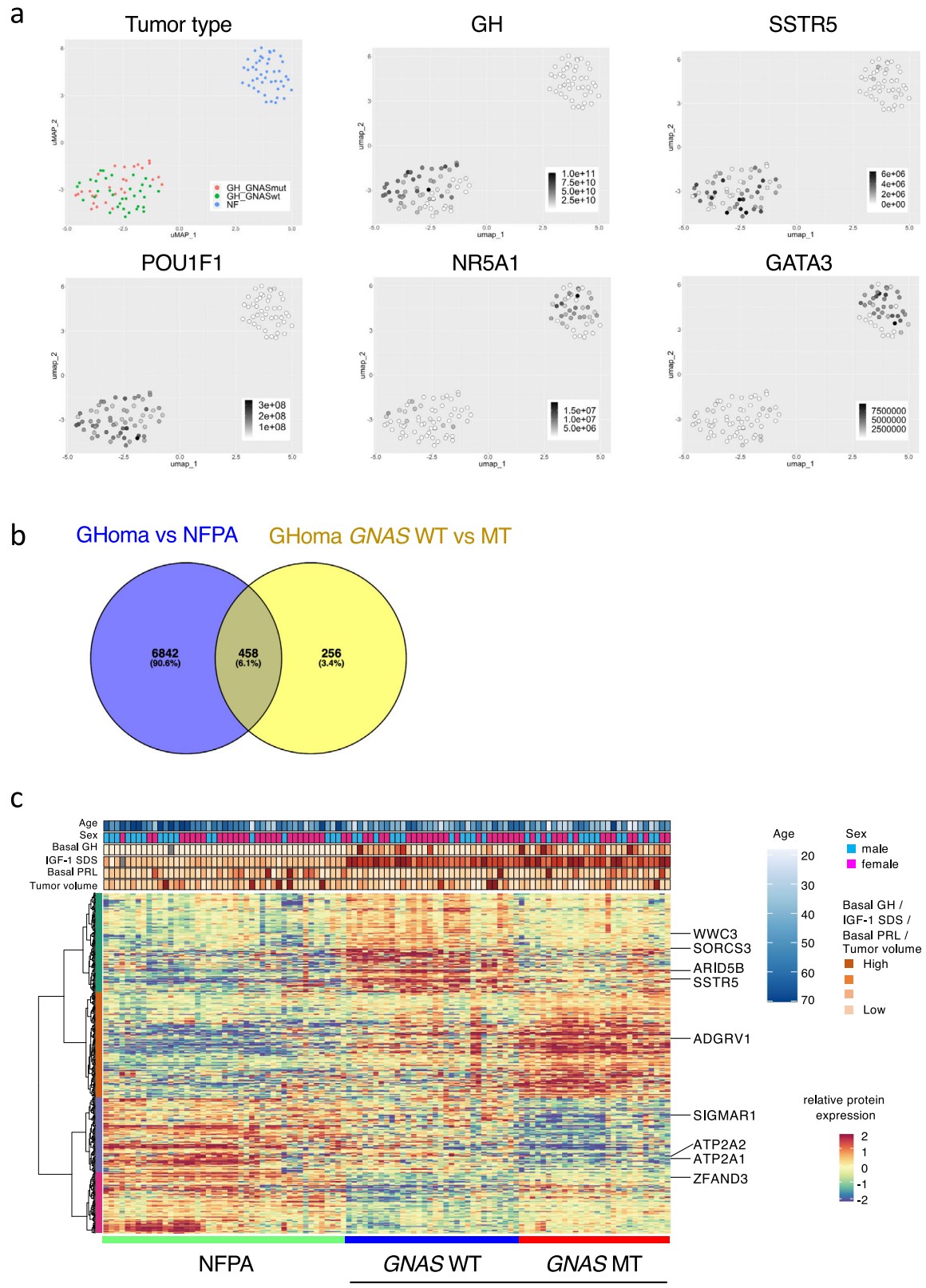

following SSA treatment. The expression levels of ADGRV1 and SORCS3 were correlated with the GH change rate following the octreotide loading test (Supplementary Fig. 4).

The GH change rate following the octreotide loading test, which has been reported to correlate with the SSTR2 mRNA expression level, did not correlate with the SSTR2 or SSTR5

protein expression levels or with the SSTR2/5 protein expression ratio (Fig. 5a–c). We then performed correlation analyses between all of the protein expression values derived from the nontargeted proteomics analysis and the GH change rate following octreotide loading test. We detected positive correlations for ATPase sarcoplasmic/endoplasmic reticulum Ca2+ transporting

**Fig. 3 Nontargeted proteomics analysis of pituitary adenomas/PitNETs. a** Uniform manifold approximation and projection (UMAP) analysis showing the clusters of NFPA and GHoma depending on tumor types: nuclear receptor subfamily 5 group A member 1 (NR5A1), GATA-binding protein 3 (GATA3), POU class 1 homeobox 1 (POU1F1), GH1, and somatostatin receptor 5 (SSTR5) using multiomics data. **b** Differential expression of proteins derived from nontargeted proteomics was estimated and analyzed with the Brunner–Munzel test. Venn diagram of overlapping differentially expressed molecules. **c** Heatmap showing differentially expressed molecules identified by proteomics analysis in NFPA and GHoma. NFPA: nonfunctional pituitary adenoma/non-functioning pituitary neuroendocrine tumor (PitNETs), GHoma: GH-producing pituitary adenoma/somatotroph Pituitary neuroendocrine tumor (PitNETs).

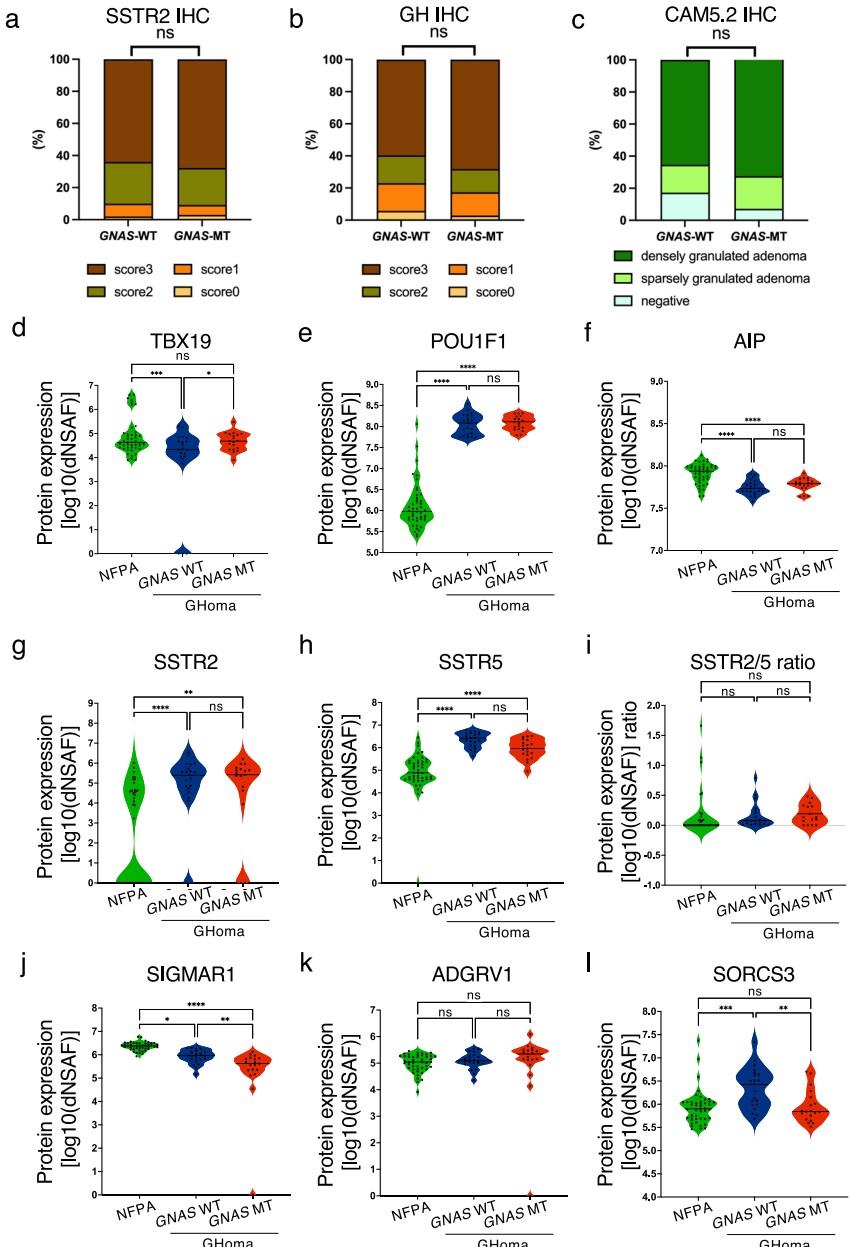

**Fig. 4 Protein expression levels in NFPAs/non-functioning PitNETs, *GNAS*-WT GHomas/somatotroph PitNETs, and *GNAS*-MT GHomas/somatotroph PitNETs.** Immunohistochemistry scores of (**a**) somatostatin receptor 2 (SSTR2) and (**b**) *GNAS*-WT and *GNAS*-mutant (MT) growth hormone (GH)-producing pituitary adenomas (%). *$p < 0.05$, by chi-square test. **c** CAM5.2 cytokeratin immunostaining pattern in the *GNAS*-WT and *GNAS*-MT groups (%). *$p < 0.05$, by chi-square test. **d** TBX19, (**e**) POU class 1 homeobox 1 (POU1F1), (**f**) aryl hydrocarbon receptor-interacting protein (AIP), (**g**) SSTR2, (**h**) SSTR5, (**j**) sigma nonopioid intracellular receptor-1 (SIGMAR1), (**k**) adhesion G protein–coupled receptor V1 (ADGRV1), and (**l**) sortilin-related VPS10 domain–containing receptor 3 (SORCS3) protein expression levels derived from nontargeted proteomics in NFPA, *GNAS*-WT GHoma and *GNAS*-MT GHoma. The protein expression values were log (base 10) transformed. **i** Protein expression ratio of SSTR2 to SSTR5 in NFPA, *GNAS*-WT GHoma, and *GNAS*-MT GHoma. NFPA: nonfunctional pituitary adenoma/non-functioning pituitary neuroendocrine tumor (PitNETs), GHoma: GH-producing pituitary adenoma/somatotroph Pituitary neuroendocrine tumor (PitNETs). *$p < 0.05$, **$p < 0.01$, ***$p < 0.001$, ****$p < 0.0001$ by 1-way ANOVA. $n = 45$ in NFPA, $n = 21$ in GHoma with *GNAS* WT, $n = 19$ in GHoma with *GNAS* MT.

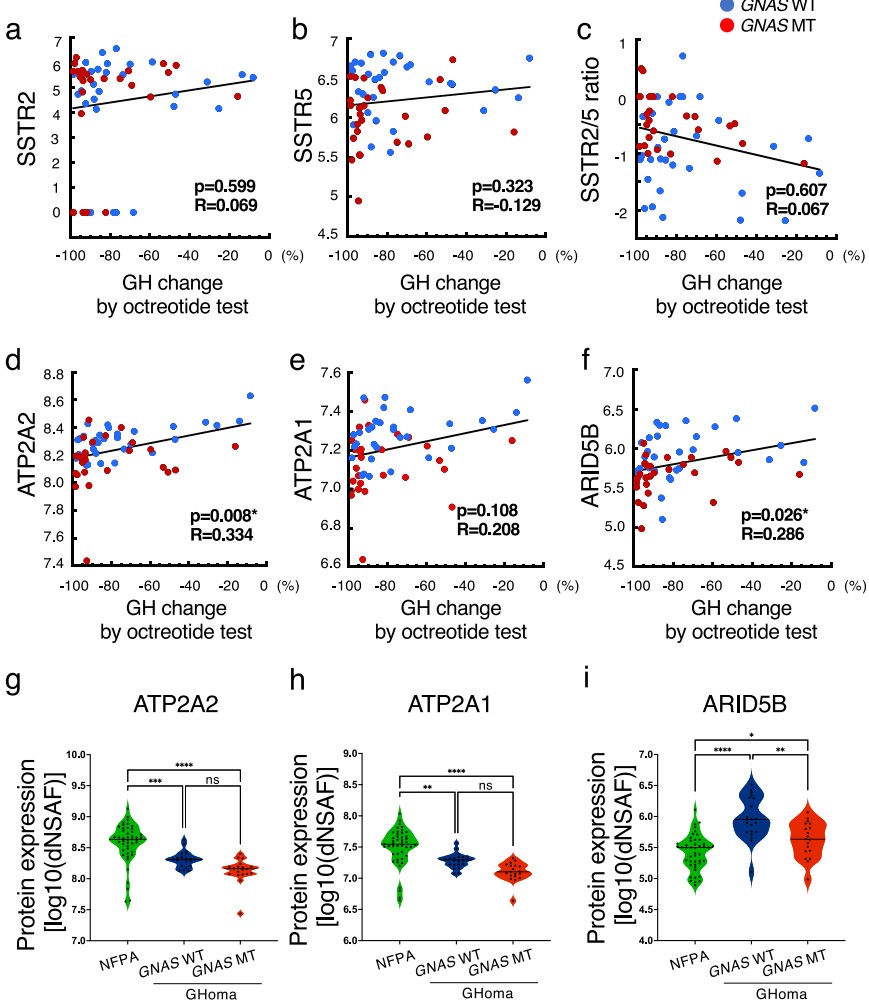

**Fig. 5 Correlation between protein expression levels and GH change rates in the octreotide loading test.** Correlations between (**a**) somatostatin receptor 2 (SSTR2) protein expression, (**b**) SSTR5 protein expression, (**c**) the SSTR2/5 ratio, (**d**) sarcoplasmic/endoplasmic reticulum calcium ATPase 2 (ATP2A2) protein expression, (**e**) ATP2A1 protein expression, and (**f**) AT-rich interaction domain 5B (ARID5B) protein expression and growth hormone (GH) change ratio by octreotide loading test (%). Data were analyzed by Pearson's correlation analysis. **g** ATP2A2, (**h**) ATP2A1, and (**i**) ARID5B protein expression levels in NFPA, *GNAS*-WT GHoma, and *GNAS*-mutant (MT) GHoma. All protein expression values were log (base 10) transformed. NFPA: nonfunctional pituitary adenoma/non-functioning pituitary neuroendocrine tumor (PitNETs), GHoma: GH-producing pituitary adenoma/somatotroph Pituitary neuroendocrine tumor (PitNETs). *$p < 0.05$, **$p < 0.01$, ***$p < 0.001$, ****$p < 0.0001$ by 1-way ANOVA. **a–f** $n = 34$ in GHoma with *GNAS* WT, $n = 26$ in GHoma with *GNAS* MT. **g–i** $n = 45$ in NFPA, $n = 21$ in GHoma with *GNAS* WT, $n = 19$ in GHoma with *GNAS* MT.

2 (ATP2A2) and AT-rich interaction domain 5B (ARID5B) with the GH change rate following the octreotide loading test, although no correlation was observed for ATPase sarcoplasmic/endoplasmic reticulum Ca2+ transporting 1 (ATP2A1) (Fig. 5d–f). The protein expression levels of ATP2A2 and ATP2A1 were significantly lower in the GHomas/somatotroph PitNETS group than in the NFPAs/non-functioning PitNETs group, and ARID5B expression was significantly higher in the GHomas/somatotroph PitNETs group. These 3 molecules were expressed at significantly lower levels in the *GNAS*-MT group than in the *GNAS*-WT group (Fig. 5g–i)

To clarify the pathology of functional pituitary adenomas/PitNETS, we performed correlation analyses between the nontargeted proteomics data and the tumor volume change rate following SSA treatment because it is important to consider not only the production and secretion of hormones but also tumorigenesis. SSTR2 expression, SSTR5 expression, and the SSTR2/5 ratio did not correlate with the tumor volume change rate following SSA treatment (Fig. 6a–c). The tumor volume change rate following SSA treatment was correlated with the expression levels of WWC family member 3 (WWC3) and serine incorporator 1 (SERINC1)

and tended to correlate with zinc finger AN1-type containing 3 (ZFAND3) expression (Fig. 6d–f). The expression of WWC3 was significantly lower in the *GNAS*-MT group than in the *GNAS*-WT group, whereas SERINC1 and ZFAND3 expression levels were significantly higher in the *GNAS*-MT group than in the *GNAS*-WT group (Fig. 6g–i).

These differentially expressed protein candidates from the nontargeted proteomics analysis were evaluated by pathological examination. Sections were stained with SIGMAR1, ATP2A2, ARID5B, WWC3, and SERINC1 (Fig. 7a–e), and IHC scoring was performed as described in the Methods. SIGMAR1 was significantly increased in the NFPAs/non-functioning PitNETs group compared with the *GNAS*-MT group. (Fig. 7f). ATP2A2 was significantly increased in the NFPAs/non-functioning PitNETs group compared with the GHomas/somatotroph PitNETs groups (Fig. 7g). ARID5B was significantly increased in the *GNAS*-WT group compared with the *GNAS*-MT group (Fig. 7h). There were no significant differences in WWC3 and SERINC1 among the three groups. (Fig. 7i, j). These expression patterns were similar to the trends observed using nontargeted proteomics analysis.

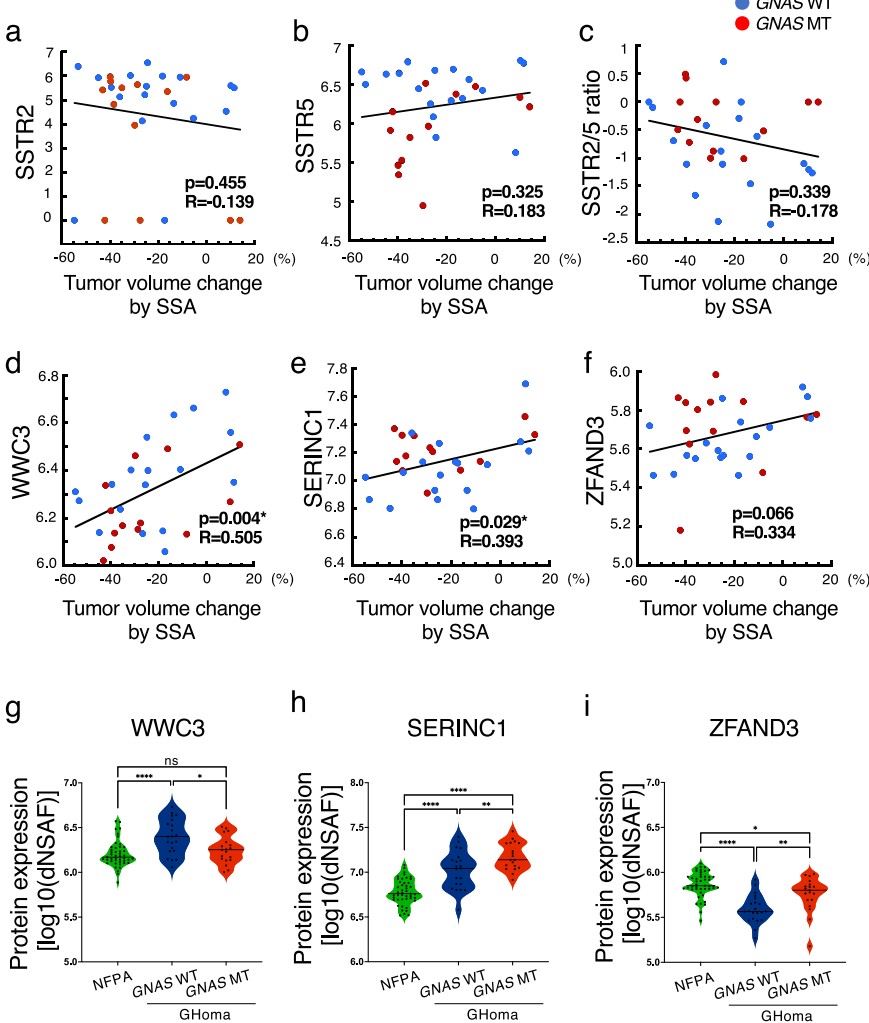

**Fig. 6 Correlation between protein expression levels and tumor volume change rate in the SSA test.** Correlations between (**a**) somatostatin receptor 2 (SSTR2) protein expression, (**b**) SSTR5 protein expression, (**c**) the SSTR2/5 ratio, (**d**) WWC family member 3 (WWC3) protein expression, (**e**) serine incorporator 1 (SERINC1) protein expression, and (**f**) zinc finger AN1-type containing 3 (ZFAND3) protein expression and the tumor volume change rate in the somatostatin analog (SSA) test (%). Data were analyzed by Pearson's correlation analysis. **g** WWC3, (**h**) SERINC1, and (**i**) ZFAND3 protein expression levels in NFPA, *GNAS*-WT GHoma, and *GNAS*- MT GHoma. All protein expression values were log (base 10) transformed. NFPA: nonfunctional pituitary adenoma/non-functioning pituitary neuroendocrine tumor (PitNETs), GHoma: GH-producing pituitary adenoma/somatotroph Pituitary neuroendocrine tumor (PitNETs). *$p < 0.05$, **$p < 0.01$, ****$p < 0.0001$ by 1-way ANOVA. **a–f** $n = 18$ in GHoma with *GNAS* WT, $n = 13$ in GHoma with *GNAS* MT. **g–i** $n = 45$ in NFPA, $n = 21$ in GHoma with *GNAS* WT, $n = 19$ in GHoma with *GNAS* MT.

## Discussion

In this study, we integrated clinical characteristics, genetic analysis, and biochemical analysis methods, including IHC and proteomics, to assess GHomas/somatotroph PitNETs in a large cohort. Using transomics classification, we showed that mRNA and protein levels were modestly correlated in pituitary adenomas/PitNETs, similar to reports for human colon and rectal cancer studies[24]. TCS analysis revealed that *GNAS* was the only driver gene of GHomas/somatotroph PitNETs, as previously reported, and the nontargeted proteomics analysis revealed that *GNAS* was an important player in shaping various acromegaly characteristics.

The present analysis revealed that 57.0% of patients had *GNAS*-MT. The prevalence of *GNAS* in GHomas/somatotroph PitNETs has a wide range, 4–59%, according to previous reports[26]. The clinical characteristics of patients with *GNAS*-MT were consistent with those in previous reports[11,12]. However, the clinical data obtained for each group presented an extremely wide range, indicating that many patients showed exceptional characteristics in some respects. Actually, among reports with low prevalence, there

are reports of significant characteristics with *GNAS*-MT[12] or no characteristic differences with or without *GNAS*-MT[26]. Next, for recurrent genes aside from *GNAS* in acromegaly, we identified 18 recurrent genes using Neou M et al.'s method. There are previous reports that the only recurrent gene is *GNAS*[1,27], while Neou M et al. identified ~30 recurrent genes[23]. This finding suggests that the clinical characteristics of acromegaly are not reflective of a single genetic mutation but are the result of several genetic events and gene expression changes, consistent with previous reports regarding the involvement of CNAs or epigenetic alterations[27]. The investigation of associations between these genetic events and the clinical phenotypes of acromegaly will be explored in future studies. Although *GNAS*-MT cannot explain all of the clinical characteristics, proteomics analysis showed that *GNAS*-MT were related to many differentially expressed proteins. Notably, GO analysis revealed the impacts of *GNAS*-MT on the GPCR pathway. GPCRs play important roles in GHomas/somatotroph PitNETs as signal transducers of GH-releasing hormone, leading to the production and secretion of GH, and as SSTRs, which regulate GH secretion

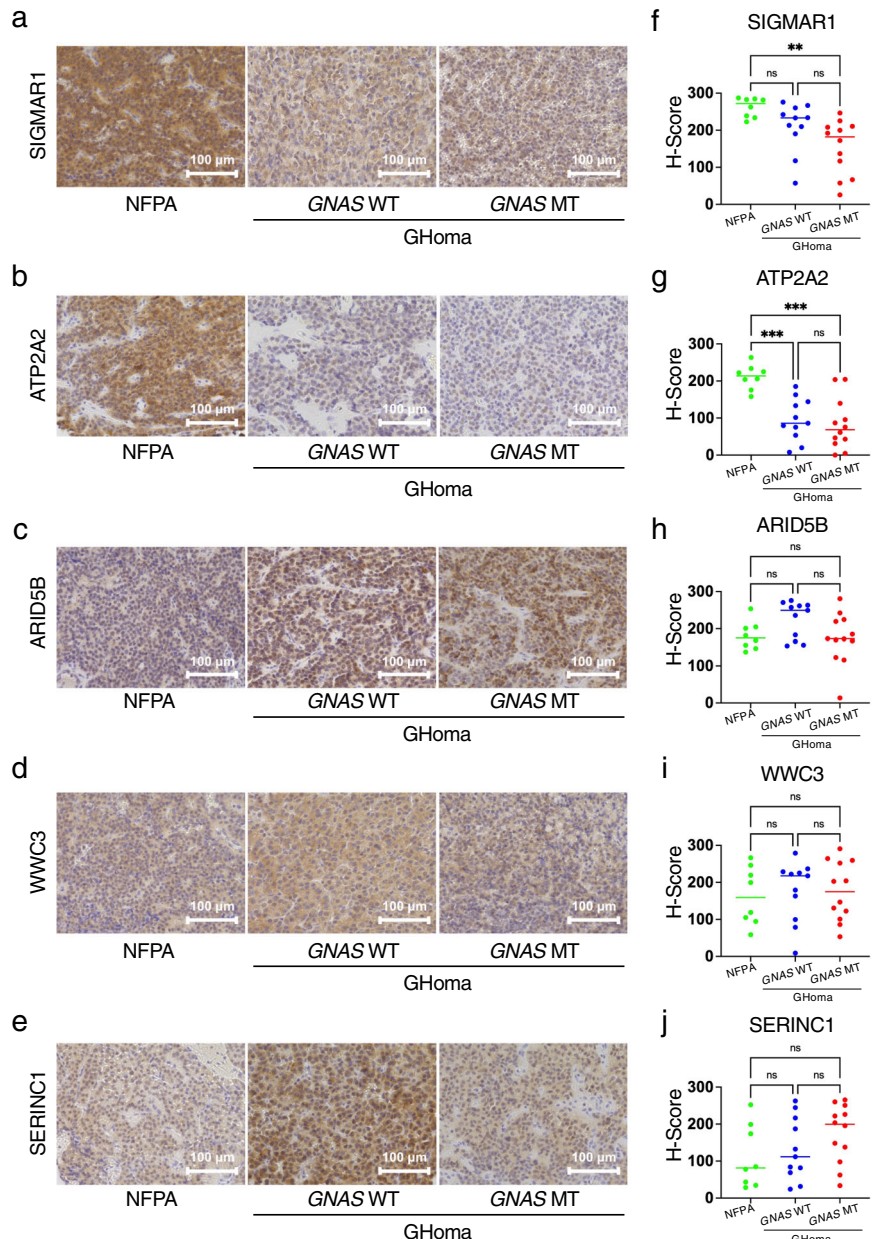

**Fig. 7 Immunohistochemistry of differentially expressed candidates in nontargeted proteomics.** Sections were stained for (**a**) sigma nonopioid intracellular receptor-1 (SIGMAR1), (**b**) sarcoplasmic/endoplasmic reticulum calcium ATPase 2 (ATP2A2), (**c**) AT-rich interaction domain 5B (ARID5B), (**d**) WWC family member 3 (WWC3), and (**e**) serine incorporator 1 (SERINC1). IHC scoring of (**f**) SIGMAR1, (**g**) ATP2A2, (**h**) ARID5B, (**i**) WWC3 and (**j**) SERINC1 was performed by modifying McCarty's H-score. NFPA: nonfunctional pituitary adenoma/non-functioning pituitary neuroendocrine tumor (PitNETs), GHoma: GH-producing pituitary adenoma/somatotroph Pituitary neuroendocrine tumor (PitNETs). **$p < 0.01$, ***$p < 0.001$ by 1-way ANOVA. $n = 8$ in NFPA, $n = 11$ in GHoma with *GNAS* WT, $n = 12$ in GHoma with *GNAS* MT.

and cell proliferation. *GNAS* mutations (*GNAS*-MT) have been proposed to trigger different acromegaly clinical characteristics through the altered expression of components in these important pathways. In particular, the induction of the GPCR pathway may contribute to differences in the GH change rate by octreotide loading test which is known to correlate with GH change rate following SSA treatment, based on the comparison of clinical phenotypes observed in our study. However, SSTR2, which is the typical GPCR expressed in GHomas/somatotroph PitNETs, was not identified as a differentially expressed molecule involved in the GPCR pathway. Conventional SSAs, such as octreotide long-acting release (LAR) or lanreotide, bind primarily with SSTR2, and some previous reports have identified differences in SSTR2 mRNA

expression in patients with and without *GNAS*-MT[28,29]. In addition, neither the SSTR2 IHC score nor the protein expression level derived from the proteomics analysis correlated with the GH change rate following octreotide test or with the tumor volume change rate following SSA treatment, although some reports showed that the SSTR2 mRNA expression level was positively correlated with the response to SSA in GHomas/somatotroph PitNETs[3,13,14]. However, our analysis was based on protein expression levels rather than mRNA expression levels, and SSTRs may be downregulated in patients preoperatively treated with SSA, although we did not identify differences in SSTR2 protein expression levels between patients with and without preoperative SSA treatment (Supplementary Fig. 5).

SIGMAR1 plays an important role in the cellular functions of various tissues associated with the endocrine, immune, and nervous systems, in addition to cancer cells. SIGMAR1 primarily functions in physiological and pathophysiological processes of the CNS, such as pain, memory, neurodegenerative diseases, stroke, and addiction. SIGMAR1 is activated in response to tissue injury and during disease development to promote cell survival. However, interactions between SIGMAR1 and ion channels may change cellular behaviors in response to the microenvironment, leading to the unexpected consequence of tumor development[30]. Although SSTR5 and SIGMAR1 are not associated with GH change by octreotide loading test and tumor volume change by preoperative SSA treatment, these molecules can regulate GH secretion in response to GPCR signals.

ATP2A2 and ARID5B were identified as being correlated with the GH change rate in response to octreotide treatment, and WWC3, SERINC1, and ZFAND3 were identified as being correlated with the tumor volume change rate in response to SSA treatment. All of these proteins demonstrated significant differences in expression between the GNAS-WT and GNAS-MT groups. ATP2A2 is an intracellular pump located in the sarcoplasmic or endoplasmic reticulum. Interestingly, a physical association with ATP2A2 has been described as a universal feature among GPCRs, which may be required for the efficient folding or membrane integration of GPCRs[31]. ARID5B forms a histone H3K9Me2 demethylase complex with PHD finger protein 2 and regulates the transcription of target genes[32]. Recently, several reports have indicated that chromatin modification genes are dysregulated in pituitary adenomas/PitNETs; for example, pituitary adenomas/PitNETs exhibit increased acetylation of H3K9[33]. WWC3 is an upstream regulator of the Hippo signaling pathway, which controls cell proliferation and organ growth[34]. SERINC1 is a membrane protein whose expression is restricted to the CNS[35]. ZFAND3 is a zinc finger protein involved in nucleic acid recognition, transcriptional activation, protein folding, and assembly and causes tumor invasion in glioblastoma[36]. Therefore, these molecules may be related to GH secretion or tumor development. In present study, pathological expression pattern was consistent with protein expression level derived from nontargeted proteomics analysis in SIGMAR1, ATP2A2 and ARID5B. Among them, ATP2A2 and ARID5B expression levels were significantly correlated with the GH change rate after the octreotide loading test, which correlated with the effect of SSA treatment. Currently, in clinical practice, SSTR2 immunostaining for surgical specimens of GHomas/somatotroph PitNETs is often performed as a predictor of the effect of additional postoperative SSA therapy. On the other hand, some reports have shown no correlation between SSTR2 pathological expression and SSA efficacy, just as we have shown above in protein expression data derived from nontargeted proteomics analysis. Our results suggested that ATP2A2 and/or ARID5B immunostaining may be a predictor of the efficacy of SSA treatment, either independently or in combination with SSTR2 immunostaining in noncurative resection. Further investigations into how these molecules are involved in the development of GHomas/somatotroph PitNETs or how they impact treatment efficacy are needed.

This study has two major limitations. First, we focused on only 36 genes by TCS; however, many other genes are likely involved in the tumorigenesis of GHomas/somatotroph PitNETs. Accordingly, we also examined the CNAs of only 36 genes. Second, we were unable to evaluate germline genomic information.

In summary, we integrated gene alteration data, proteomics data, and clinical information from a large cohort of patients with GHomas/somatotroph PitNETs. GNAS mutational status could not provide a complete explanation of the clinical characteristics of acromegaly; however, proteomics analysis showed that GNAS-

MT influenced the expression of many proteins, including those involved in the GPCR pathway. We confirmed that these molecules were important factors underlying the clinical and biochemical features involved in the responsiveness of GHomas/somatotroph PitNETs to medical treatment.

## Methods

**Subjects.** We studied 121 patients who underwent TSS for sporadic acromegaly at Toranomon Hospital between 2013 and 2019 and studied 45 patients who were diagnosed with NFPAs/non-functioning PitNETs. The acromegaly diagnosis was based on typical symptoms, such as a characteristic appearance and the enlargement of the limbs and tongue, in addition to laboratory findings, including elevated basal GH, sex- and age-adjusted IGF-1, and unsuppressed GH after an OGTT, with MRI evidence of pituitary adenomas/PitNETs. GH immunoreactivity was confirmed histologically in all samples by a pathologist. Based on MRI, the adenoma volume was calculated as $0.5 \times$ width $\times$ length $\times$ height (mm$^3$)[37], and the Knosp grade was used to evaluate invasiveness to the cavernous sinus[38]. We defined the GH reduction rate after preoperative SSA as follows: $100 \times$ (serum GH after preoperative SSA treatment − serum basal GH)/serum basal GH. Similarly, the tumor volume reduction rate after preoperative SSA treatment was calculated as follows: $100 \times$ (tumor volume after preoperative SSA treatment − tumor volume)/serum basal GH. Either octreotide LAR or lanreotide was used as the preoperative SSA. We defined postoperative remission as (1) nadir GH < 0.4 ng/mL after OGTT, (2) normalization of age- and sex-adjusted IGF-1, and (3) improvement of symptoms reflecting disease activity, such as headache or hyperhidrosis.

Serum GH levels were measured by electrochemiluminescence immunoassay using an EClusys kit (Roche Diagnostics, Tokyo, Japan), and serum IGF-1 levels were measured by an immunoradiometric assay using a 'Daiichi' IGF-2 IRMA kit (FUJIREBIO Inc. Tokyo, Japan).

**Targeted capture sequencing and copy number analysis.** TCS was performed using a HiSeq 2500 (Illumina). We chose 36 genes for TCS according to three criteria: (1) genes related to GHomas/somatotroph PitNETs according to the Catalogue of Somatic Mutations in Cancer (COSMIC); (2) genes related to SSA treatment resistance and tumor progression in GHomas/somatotroph PitNETs; and (3) genes known to be associated with cancer progression. All 36 genes are listed in Supplemental Data 2. Sequence reads were aligned to the human reference genome (GRCh38). Mutation calling was performed using Mutect2, VarSCan2, and LoFreq. We determined candidate mutations according to four parameters: (1) those with variant allele frequency values ≥0.01 in tumors were retained; (2) those with minor allele frequency values ≥0.001 according to The Exome Aggregation Consortium were excluded; (3) those with synonymous single-nucleotide variants (SNVs) were excluded; and (4) those with SNVs in introns were excluded. The pathogenicity of the variants was initially screened using COSMIC (https://cancer.sanger.ac.uk/cosmic). COSMIC-negative variants were then assessed with Mutation Taster (http://www.mutationtaster.org/), PolyPhen2 (http://genetics.bwh.harvard.edu/pph2/dbsearch.shtml) and fathmmMKL (http://fathmm.biocompute.org.uk/fathmmMKL.htm#download). We defined candidate mutations as pathogenic when pathogenic variants were detected by at least two software programs.

Genome-wide CNAs were analyzed by Python 2.6.6 using CNVkit library version 0.9.1 with default parameters from NGS data[39]. Sequencing coverage of targeted regions in all samples was assessed and used to create pooled reference data. CNAs were defined as deletions if the log$_2$ copy ratio was reported to be below −1 and as amplifications if the log$_2$ copy ratio was above 1.

**Nontargeted proteomics analysis.** Proteins were isolated from tumor samples using phenol-guanidinium isothiocyanate (P/GTC) reagent according to the manufacturer's protocol[40]. Proteins were precipitated from the phenol/ethanol phase by the addition of acetone. The protein extract was digested, and the peptides were analyzed by liquid chromatography tandem mass spectrometry (LC-MS/MS; Orbitrap Exploris 480).

**RNA sequencing analysis.** Total RNA was extracted from all tumor tissues using P/GTC. The RNA-seq library was prepared according to the manufacturer's protocol using the QuantSeq 3′mRNA-Seq Library Prep Kit (Lexogen, Vienna, Austria) and sequenced using a NextSeq 500 sequencer (Illumina, San Diego, CA). The default parameters of TopHat2 version 2.0.8 and Bowtie2 version 2.1.0 were used to map sequence reads to the human genome 38, and gene annotation information was provided by NCBI. The Cufflinks software tool was used to estimate text abundance (version 2.1.1). Cufflinks was run with the same reference annotation as TopHat2 to generate the fragments read per kilobase per million mapped reads (FPKM) values of the KNOWN gene model. We used an FPKM cutoff of 1 to identify an expressed gene.

**Immunohistochemistry.** Surgically removed adenoma tissues were evaluated by pathological and immunohistochemical examinations using the following antibodies: GH (Dako, Carpinteria, CA, USA; A0570, 1:10), cytokeratin (CK; CAM 5.2;

BD Biosciences, San Jose, CA, USA; 345779, 1:10), SSTR2a (Abcam, Cambridge, UK, ab134152, 1:1000), SIGMAR1 (Sigma-Aldrich, Tokyo, Japan, HPA018002, 1:100), ATP2A2 (Sigma-Aldrich, Tokyo, Japan, HPA062605, 1:300), ARID5B (Sigma-Aldrich, Tokyo, Japan, HPA015037, 1:100), WWC3 (Sigma-Aldrich, Tokyo, Japan, HPA039814, 1:500), and SERINC1 (Sigma-Aldrich, Tokyo, Japan, HPA035738, 1:50). The immunostaining results for GH were classified into four grades according to the number of immunopositive cells: 0, no staining; 1, <30%; 2, 30%–70%; and 3, >70% staining of all tumor cells. The SSTR2 protein expression level was scored as previously reported[41]. Adenomas were categorized as densely granulated adenomas or sparsely granulated adenomas according to CAM5.2 cytokeratin immunostaining, as previously described[42]. These IHC evaluations were performed by one experienced pathologist and one researcher.

IHC scoring of SIGMAR1, ATP2A2, ARID5B, WWC3, and SERINC1 was performed by two researchers independently by modifying McCarty's H-score[43], which integrated the intensity and frequency of staining, in 23 acromegaly patients and 8 NFPAs/non-functioning PitNETs patients. First, 4 hot spots were selected in each section with low magnification and then the percentage of immunoreactive cells was counted with high magnification in each hot spot. The staining intensity was evaluated in each hot spot as follows: 0 no staining; 1+ weakly stained; 2+ moderately stained; 3+ strongly stained. The IHC score in each hot spot was calculated by the staining intensity score $(0–3) \times$ percentage of stained cells, and then the average of the IHC scores for the 4 hot spot scores was calculated. The average of IHC score of the 2 researchers was taken as the final score.

**Statistics and reproducibility**. Normally distributed data are expressed as the mean ± SD, whereas nonnormally distributed data are expressed as the median (interquartile range; IQR). Differences between two groups were determined by the chi-square test or Fisher's exact test for categorical variables, by a two-sample t test for normally distributed continuous variables, or by the Mann–Whitney U test for nonnormally distributed continuous variables. Continuous variables in multiple groups were compared by one-way ANOVA for normally distributed variables or by the Kruskal–Wallis test for nonnormally distributed variables. Correlations were assessed using Pearson's product-moment correlation coefficients. We considered differences with $p$ values < 0.05 to be significant. The differential expression of proteins derived from nontargeted proteomics analysis was estimated and analyzed with the Brunner–Munzel test. Statistical analyses were performed in SPSS 2 for Windows (SPSS Japan Inc. Tokyo) and R. Because unique human pituitary adenomas/PitNETs samples were used in this study and no cell lines, primary culture or research animals, replication was not possible.

**Study approval**. The collection of adenoma samples and patient information was approved by the ethics committee of Chiba University Graduate School of Medicine and Toranomon Hospital. Written informed consent was received from participants before inclusion in the study.

**Reporting summary**. Further information on research design is available in the Nature Portfolio Reporting Summary linked to this article.

## Data availability

The raw and processed RNA sequencing data generated in this study have been deposited in the Gene Expression Omnibus (GEO) database under accession code GSE213527. The mass spectrometry proteomics data have been deposited in the ProteomeXchange Consortium (http://proteomecentral.proteomexchange.org) via the jPOST partner repository (http://jpostdb.org) with the dataset identifier PXD036604. Source data are provided with this paper in Supplementary Data 6.

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

## Acknowledgements

This work was supported by grants from the Ministry of Education, Culture, Sports, Science and Technology (Japan) [Grants-in-Aid: for Scientific Research (S) 26221305, JP19H05650, (B) #21H02974, #19H03708, #22300325; (C) #21K07145, #21K08524, #20K08397, #20K07561, #19K07635, 19K08972, #18K07439, #18K08464; Challenging Research (Exploratory) #21K19398, #20K21837; Early-Career Scientists #20K17527]. T. Tanaka was supported by Japan Society for the Promotion of Science KAKENHI grant JP19H03708. This work was partly supported by The Uehara Memorial Foundation, Mochida Memorial Foundation for Medical and Pharmaceutical Research, The Naito Foundation, Mitsui Life Social Welfare Foundation, Princes Takamatsu Cancer Research Fund, Takeda Science Foundation, Senshin Medical Research Foundation, Kose Cosmetology Research Foundation, Japan Diabetes Foundation, Yamaguchi Endocrine Research Foundation, The Cell Science Research Foundation, The Ichiro Kanehara Foundation for the Promotion of Medical Sciences and Medical Care, the Yasuda Memorial Medical Foundation, MSD Life Science Foundation, The Hamaguchi Foundation for the Advancement of Biochemistry, The Novartis Foundation (Japan) for Promotion of Science and the Medical Institute of Bioregulation Kyushu University Cooperative Research Project Program.

## Author contributions

A.Y., H.N.a, and T.T. designed the study. K.H., Y.I., N.F., H.N.i, and S.Y. collected the specimens. A.Y., Y.G.a, T.M., S.N.Z., T.K., Y.T., N.F., and M.N. collected clinical information and conducted experiments. O.O. directed genomic analysis and proteomics. A.Y., H.N.a, M.F., B.R., T.I., and E.K. performed bioinformatics and statistical analyses. Y.N. and N.I. performed immunohistochemistry. Y.G.o, X.S., N.I., and A.Y. performed histopathologic analyses. A.Y., H.N.a, A.K., K.Y., and T.T. analyzed, discussed, and interpreted the data. A.Y., H.N.a, N.H., M.Y., K.Y., and T.T. coordinated and directed the project and wrote the paper. All authors approved the submitted paper.

## Competing interests

The authors declare no competing interests.
