## [Peer Review File · Communications Biology]

Reviewers' comments:

Reviewer #1 (Remarks to the Author):

The study integrates the genetic alterations, protein expression profiles, transcriptomes, and clinical characteristics of GH-producing pituitary adenomas to detect molecules associated with acromegaly characteristics. The transomics classification showed that mRNA and protein levels were modestly correlated in pituitary adenoma. The TCS analysis revealed that GNAS was the only driver gene of GH-producing pituitary adenoma, and bioinformatics analysis showed the impacts of GNAS mutation on the GPCR pathway. In addition, this manuscript confirmed that GNAS mutations and novel molecules were important factors underlying the clinical and biochemical features involved in the responsiveness of GH-producing pituitary adenomas to medical treatment. Although this is an interesting study and the paper is of potential interest in the general field of pituitary research, we still have some concerns.

1. The readers may have curiosity about the clinical characteristics of the patients with GNAS mutations more detailed. For example, the characteristics of the GH-producing pituitary adenomas including extent of surgical resection and biochemical remission. And it's better to provide the follow-up data and the status of recurrence.
2. P13, L209. Reference to Figure 4h seems SSTR5 expression was higher in the GH-producing pituitary adenoma group. Please clarify and/or correct.
3. P13, L209. Reference to Figure 4j seems SIGMAR1 expression was lower in the GH-producing pituitary adenoma group. Please clarify and/or correct.
4. Please check through all the figure labels, it should be of the same font and size in all figures. e.g., Figure 6a, b, c and Figure 6 d, e, f.
5. P14, L232. Why the authors focus on three genes, WWC, SERINC1, and ZFANDS. please clarify briefly.

Reviewer #2 (Remarks to the Author):

In this study genetic, proteomic and transcriptomic analysis, in GH-producing pituitary adenomas was done.

- In results: How did you define recurrent mutations? it is not clear in the figure 1a.
- Were the number and type of mutations in GNAS analyzed in relation to clinical behavior?
- The information presented in Figure 2 is not entirely clear. Apparently there is confusion between Figure 2 and Supplemental Figure 2.
- In figure 2b what do the colors mean?
- In my opinion supplemental figure 3 could be in figure 3 and 3a could be supplemental.
- Figure 4 a-c is not well marked. What does score 0-3 mean, the bars do not correspond to the score frames.
- In the text, the SSTR2/5 molecule is missing
- the graph shows that SSTR5 is lower in the NFPA group (fig 4h)
- SIGMAR1 is more expressed in NFPA; ADGRV1 and SORCS3 expressions are no significantly different between groups.
- I suggest reviewing the following posts in order to enrich the discussion:
Giselle Fernandes Taboada Prevalence of gsp oncogene in somatotropinomas and clinically non-functioning pituitary adenomas: our experience Pituitary. 2009;12(3):165-9; Cristina L Ronchi Landscape of somatic mutations in sporadic GH-secreting pituitary adenomas Eur J Endocrinol. 2016 Mar;174(3):363-72; Peculis R. Large Scale Molecular Studies of Pituitary Neuroendocrine Tumors: Novel Markers, Mechanisms and Translational Perspectives. Cancers (Basel) 2021 Mar 19;13(6):1395.

Response to reviewers

We are very grateful for the interest and the thoughtful questions from the reviewers of our manuscript. We feel that the changes included in the revised manuscript in response to those recommendations have made it stronger and we are grateful for the encouragement to include these data. We hope that the reviewers agree. For a broad readership, Figure 7 and Supplemental Figure 5 were merged to restructure a new Figure 7, and Supplemental Figure 5 was deleted.

In the following, we respond point-by-point to the reviewers' comments.

Reviewer #1:

Comments:

The study integrates the genetic alterations, protein expression profiles, transcriptomes, and clinical characteristics of GH-producing pituitary adenomas to detect molecules associated with acromegaly characteristics. The transcriptomics classification showed that mRNA and protein levels were modestly correlated in pituitary adenoma. The TCS analysis revealed that GNAS was the only driver gene of GH-producing pituitary adenoma, and bioinformatics analysis showed the impacts of GNAS mutation on the GPCR pathway. In addition, this manuscript confirmed that GNAS mutations and novel molecules were important factors underlying the clinical and biochemical features involved in the responsiveness of GH-producing pituitary adenomas to medical treatment. Although this is an interesting study and the paper is of potential interest in the general field of pituitary research, we still have some concerns.

Reply: First, we are delighted that the reviewer had interest in our findings and in this positive assessment of our work. We are thankful for the very thoughtful comments.

1. The readers may have curiosity about the clinical characteristics of the patients with GNAS mutations more detailed. For example, the characteristics of the GH-producing pituitary adenomas including extent of surgical resection and biochemical remission. And it's better to provide the follow-up data and the status of recurrence.

Reply: Thank you so much for the important suggestion. Unfortunately, data on the extent of surgical resection were not available. However, OGTT was performed in all postoperative cases to determine biochemical remission. We collected data on biochemical remission. The mean nadir values of GH in the postoperative OGTT were 0.50 (0.26–1.00) ng/mL in *GNAS* WT and 0.39 (0.27–0.67) ng/mL in *GNAS* MT. The mean postoperative IGF-1 SDS was 0.9 ± 1.7 in *GNAS* WT and 0.5 ± 1.5 in *GNAS* MT. There were no significant differences between *GNAS* WT and MT.

When biochemical remission was defined as the criteria for remission in the OGTT, 100 out of 121 cases showed remission, 43 cases (82.7%) in *GNAS* WT, and 57 cases (82.6%) in *GNAS* MT. No significant difference was observed depending on the presence or absence of *GNAS* mutations.

Postoperative OGTT, IGF-1, IGF-1 SDS, and remission data were also added to Table 1 and Page 6 Lines 78-80. Regarding the follow-up and the status of recurrence, at least in this cohort, no patient underwent repeat surgery after more than 1 year. However, another study with a longer follow-up period is necessary to evaluate the status of recurrence precisely.

2. P13, L209. Reference to Figure 4h seems SSTR5 expression was higher in the GH-producing pituitary adenoma group. Please clarify and/or correct.

Reply: We would like to apologize for this error in the original manuscript. SSTR5 expression was higher in the GH-producing pituitary adenoma group than in the NFPA group, and there was no difference in expression with or without *GNAS* mutations. We have corrected this error (Page 14, Lines 216-218).

3. P13, L209. Reference to Figure 4j seems SIGMAR1 expression was lower in the GH-producing pituitary adenoma group. Please clarify and/or correct.

Reply: We would like to apologize for this error in the original manuscript. SIGMAR1 expression was lower in the GH-producing pituitary adenoma group than in the NFPA and was lower in *GNAS*-MT expression in the GH-producing pituitary adenoma group than in the *GNAS*-WT group. We have corrected it and clarified the descriptions of ADGRV1 and SORCS3. We have described these results in the revised manuscript (Page 14, Lines 221-227).

4. Please check through all the figure labels, it should be of the same font and size in all figures. e.g., Figure 6a, b, c and Figure 6 d, e, f.

Reply: Thank you for the important suggestions. We have checked and corrected the font and size in the figures.

5. P14, L232. Why the authors focus on three genes, WWC, SERINC1, and ZFANDS. please clarify briefly.

Reply: Thank you so much for this important question. These molecules had a stronger correlation among the molecules that correlated with the tumor volume change rate following SSA treatment compared to other molecules. To clarify the pathology of functional pituitary adenomas, we believe it is important to consider not only the production and secretion of hormones but also tumorigenesis. Interestingly, these molecules did not correlate with the inhibitory effects of SSA on GH. Therefore, there may be molecules that have some effect on tumorigenesis in GH-producing pituitary adenoma. We have described these results in the revised manuscript (Page 15, Lines 246-249).

Reviewer #2:

In this study genetic, proteomic and transcriptomic analysis, in GH-producing pituitary adenomas was done.

- In results: How did you define recurrent mutations? it is not clear in the figure 1a.

Reply: Thank you very much for raising this issue. With reference to the report by Neou M et al. (Neou M, *et al.* Pangenomic Classification of Pituitary Neuroendocrine Tumors. *Cancer Cell* **37**, 123-134 e125 (2020).), the recurrent mutations were defined as follows. Mutation calling was performed using Mutect2, VarScan2, and LoFreq. We determined candidate mutations according to four parameters: 1) those with variant allele frequency (VAF) values

≥ 0.01 in tumors were retained; 2) those with minor allele frequency (MAF) values ≥ 0.001 according to The Exome Aggregation Consortium were excluded; 3) those with synonymous single-nucleotide variants (SNVs) were excluded; and 4) those with SNVs in introns were excluded. The pathogenicity of the variants was initially screened using COSMIC. COSMIC-negative variants were then assessed with Mutation Taster, PolyPhen2 and fathmmMKL. We defined candidate mutations as pathogenic when pathogenic variants were detected by at least two software programs.

In addition to GNAS, Neou M et al. identified approximately 30 genes as recurrent genes. Using this method as a reference, we detected 18 genes as recurrent genes. No genes were found to overlap with their reports. The reason may be the one stated below. They are analyzing exome and RNA sequencing. On the other hand, we searched the TCS using a limited number of 36 genes. It is known that the TCS is more sensitive than the exome due to the difference in depth, and we believe that the difference in sequence method is related to the difference in results.

- *Were the number and type of mutations in GNAS analyzed in relation to clinical behavior?*

Reply: We would like to apologize for this mistake in the original manuscript. The number of patients with *GNAS* mutations was 69. As described in the manuscript, p. Arg201Cys was identified in 45 patients, p. Arg201His in 5 patients, p. Arg201Ser in 3 patients, p. Gln227Leu in 13 patients, p. Gln227Arg in 2 patients, and p. Gln227Glu in 1 patient. One patient carried both p. Arg201Cys and p. Gln870Leu mutations. Therefore, there were 68 patients with mutations of the locus at 201 or 227. In addition, we detected 3 novel *GNAS* mutations (p. Gly49Arg, p. Ser111Asn, and p. Ala249Asp). The number of patients was one, and we found that only p. Gly49 was located in the *GNAS* functional domain. We defined that this mutation was pathogenic. Therefore, the number of patients with *GNAS* mutations was 69. We have corrected this error from 68 to 69 and modified some descriptions (Page 6 Lines 87).

Additionally, we examined the clinical behavior among mutations in 69 patients with *GNAS* mutations. We compared 52 patients with mutations at the R201 locus and 15 patients with mutations at the Q227 locus and excluded 2 patients with duplicate mutations or mutations at the G49R locus. There were no factors showing a biologically significant difference other than a tendency to be significantly younger in the Q227 locus mutation. Because the number of Q227 locus mutations is small, the other study is necessary to confirm the significance of these findings. Please refer to the "table for reviewer #2", which summarizes these data.

- *The information presented in Figure 2 is not entirely clear. Apparently there is confusion between Figure 2 and Supplemental Figure 2.*

Reply: Thank you very much for raising this issue. To clarify the meaning of the transomics analysis, we examined the correlation between RNA and protein expression from RNA sequencing and proteomics data. Figure 2a shows a scatter plot of the correlation coefficient of a significantly positively correlated gene. Figure 2b shows the whole-gene correlation analysis. This is similar to the results of the correlation analysis of RNA sequencing and proteomics in human rectal colon cancer (Zhang B, *et al.* Proteogenomic characterization of human colon and rectal cancer. *Nature* **513**, 382-387 (2014). and may represent tumor characteristics in pituitary tumors.

We revealed that there were uncorrelated genes and negatively correlated genes. These results suggest that there are networks that could only be identified by transomics. On the other hand, the ontology related to the function of the pituitary gland was identified by

KEGG, and it was confirmed that these transomics data represent the characteristics of pituitary tumors (Figure 2c). Next, consensus clustering was used to perform group classification by RNA sequencing, proteomics, and combined transcriptome analysis. The results are shown in Supplemental Figure 2.

The appropriate cluster sizes were determined. Figure 2d shows that transomics better reflects protein expression classification than RNA.

When comparing the expression of specific transcription factors in pituitary tumors, the expression patterns of RNA and protein were similar between these molecules (Figure 2e and 2f). However, transomics suggests that there may be new molecules that exhibit significant variation in only protein expression that reflect the characteristics of pituitary adenomas (Figure 2g). Based on the results of the bioinformatics analyses shown in Figure 2, we focused on both proteomics and genomics approaches for the characterization of GH-producing pituitary adenomas.

We have described these results in the revised manuscript (Page 9, Lines 128-130, 131-133, 136-137, 141-142, Page 10, Lines 151-152, 158-160).

- In figure 2b what do the colors mean?

Reply: We apologize for lack of description about what the colors mean in figure legend. Negative correlations are shown in green and positive correlations in red. We added the description in the legend of Figure 2b.

- In my opinion supplemental figure 3 could be in figure 3 and 3a could be supplemental.

Reply: Thank you for this good suggestion. We have changed Figure 3 and Supplemental Figure 3 and their legends (Page 34, Lines 642-646). According to your suggestion, the explanation of the data in Figure 3 has been improved.

- Figure 4 a-c is not well marked. What does score 0-3 mean, the bars do not correspond to the score frames.

Reply: We apologize for these errors in the original figures. First, the score frame has been changed to color gradation to match the frame of Figure 1a. Second, the immunostaining results for GH were divided into four grades according to the number of immunopositive cells: 0, no staining; 1, <30%; 2, 30%-70%; and 3, >70% staining of all tumor cells. The SSTR2 protein expression level was scored as follows: score 0: absence of immunoreactivity; score 1: pure cytoplasmic immunoreactivity, either focal or diffuse; score 2: membranous reactivity in less than 50% of tumor cells, irrespective of the presence of cytoplasmic staining; score 3: circumferential membranous reactivity in more than 50% of tumor cells, irrespective of the presence of cytoplasmic staining (Volante M, *et al.* Somatostatin receptor type 2A immunohistochemistry in neuroendocrine tumors: a proposal of scoring system correlated with somatostatin receptor scintigraphy. *Mod Pathol* **20**, 1172-1182 (2007)). Adenomas were categorized as densely granulated adenomas or sparsely granulated adenomas according to CAM5.2 cytokeratin immunostaining.

- In the text, the SSTR2/5 molecule is missing

Reply: We would like to apologize for this error in the original manuscript. The SSTR2/5 protein expression ratio was not significantly different between the NFPA and GH-producing

pituitary adenoma groups, nor were they expressed with or without *GNAS* mutations in the GH-producing pituitary adenoma group. We have corrected this error (Page 14, Lines 218-221).

- the graph shows that SSTR5 is lower in the NFPA group (fig 4h)

Reply: We would like to apologize for this error in the original manuscript. *SSTR5* expression was higher in the GH-producing pituitary adenoma group than in the NFPA group, and there was no difference in expression with or without *GNAS* mutations. We have corrected this error (Page 14, Lines 216-218).

- SIGMAR1 is more expressed in NFPA; ADGRV1 and SORCS3 expressions are no significantly different between groups.

Reply: We apologize for these errors in the original manuscript. *SIGMAR1* expression was lower in the GH-producing pituitary adenoma group than in the NFPA group, and *ADGRV1* expression was not significantly different between the NFPA and GH-producing pituitary adenoma groups. *SORCS3* expression was higher in GH-producing pituitary adenoma *GNAS*-WT than in NFPA, but there was no significant difference between NFPA and *GNAS*-MT (Figure 4i). We have corrected them and clarified the descriptions of *SIGMAR1*, *ADGRV1*, and *SORCS3*. We have described these results in the revised manuscript (Page 14, Lines 221-227).

*- I suggest reviewing the following posts in order to enrich the discussion:
Giselle Fernandes Taboada Prevalence of gsp oncogene in somatotropinomas and clinically non-functioning pituitary adenomas: our experience Pituitary. 2009;12(3):165-9; Cristina L Ronchi Landscape of somatic mutations in sporadic GH-secreting pituitary adenomas Eur J Endocrinol. 2016 Mar;174(3):363-72; Peculis R. Large Scale Molecular Studies of Pituitary Neuroendocrine Tumors: Novel Markers, Mechanisms and Translational Perspectives. Cancers (Basel) 2021 Mar 19;13(6):1395.*

Reply: Thank you for this excellent suggestion. There are some reports that the recurrent gene has been concluded to be only *GNAS*. On the other hand, Neon and Assie et al. have examined pathogenicity from databases and identified recurrent genes. In our analysis, we can say that the sequence is highly sensitive because we have a sufficient depth in the part where we are using TCS. In addition, multiomics analysis of RNA sequencing and proteomics suggested that there may be a molecular network similar to cancer in pituitary tumors. Unfortunately, the recurrent genes we identified were unable to be characterized clinically. However, *GNAS* mutations alone have sufficient potential to contribute to the elucidation of the mechanisms that contribute to the pathogenesis of GH-producing pituitary adenoma. We have discussed these points in the revised manuscript (Page 17, Lines 274-275, 278-283).

We hope that these responses have satisfactorily addressed the thoughtful comments of the reviewers. We are very grateful for their many points of encouragement to improve this paper and are delighted that the data have added strength to the case for the concepts we describe. We hope that the editors will now find this work acceptable for publication in *Communications Biology*.

Table for reviewer #2. Comparison of clinical behavior between patients with GNAS R201 locus mutations and GNAS Q227 mutations.

	All GNAS mutant patients (n=69)	R201 mutant (n=52)	Q227 mutant (n=15)	p value
clinical characteristics				
Age (years)	49.0 (38.5-59.5)	51.1 ± 14.4	40.1 ± 14.6	0.012
Sex, n (male/female)	31/38	24/28	7/8	0.972
Basal GH(ng/mL)	16.9 (8.3-48.5)	13.5 (8.11-36.6)	19.1 (10.0-68.9)	0.383
IGF-1 (ng/mL)	566.0 (505.0-817.5)	635.9 ± 220.3	763.2 ± 262.2	0.064
IGF-1 SDS	6.9±2.3	6.9 ±2.1	7.3 ± 2.6	0.581
PRL (ng/mL)	20.2 (9.8-37.3)	20.3 (8.5-42.3)	17.8 (13.6-27.4)	0.735
GH change by octreotide test (%)	-89.6 (-95.0- -69.6)	-91.3 (-95.0- -78.9)	-91.3 (-93.4- -80.1)	0.561
GH change by bromocriptine test (%)	-79.1 (-90.2- -55.4)	-79.3 (-89.3- -47.4)	-68.7 (-87.1- -32.3)	0.440
Tumor volume (mm ³)	1135.4 (406.2-3493.1)	908.0 (395.5-2880.5)	2175.0 (998.5-4633.5)	0.116
Knosp grade 0-2/3-4, n	60/9	47/5	11/4	0.088
preoperative therapy				
preoperative SSA treatment, n (%)	34 (49.3%)	23 (44.2%)	9 (60.0%)	0.281
GH change by preoperative SSA (%)	-83.2 (-92.7 - -64.2)	-84.3 (-92.9- -78.9)	-81.7 (-86.4 - -56.4)	0.246
Tumor volume change by preoperative SSA (%)	-26.2±23.5	-27.6 ± 23.6	-24.4 ± 24.6	0.738
postoperative profile				
postoperative treatment, n(%)	12 (17.9%)	9 (17.3%)	3 (20.0%)	0.811

GH, Growth Hormone; PRL, Prolactine; SSA, somatostatin analog

REVIEWERS' COMMENTS:

Reviewer #1 (Remarks to the Author):

The "response to reviewers" has clearly clarified the confusing questions of this manuscript. I recommend the manuscript can be considered for acceptance.

Reviewer #2 (Remarks to the Author):

The questions and observations made to the authors were satisfactorily resolved.